# Characterization of Amnion-Derived Membrane for Clinical Wound Applications

**DOI:** 10.3390/bioengineering11100953

**Published:** 2024-09-24

**Authors:** Alison L. Ingraldi, Tim Allen, Joseph N. Tinghitella, William C. Merritt, Timothy Becker, Aaron J. Tabor

**Affiliations:** 1Axolotl Biologix, Scottsdale, AZ 85260, USA; aingraldi@axobio.com (A.L.I.);; 2Biological Sciences, Northern Arizona University, Flagstaff, AZ 86011, USA; jnt233@nau.edu; 3Mechanical Engineering and Center for Materials Interfaces in Research and Applications (MIRA), Northern Arizona University, Flagstaff, AZ 86011, USA; wmerritt@aneuvas.com (W.C.M.);

**Keywords:** human amniotic membrane (hAM), dual-layered dehydrated amnion–amnion (dHAAM), cellular biocompatibility, mechanical properties

## Abstract

Human amniotic membrane (hAM), the innermost placental layer, has unique properties that allow for a multitude of clinical applications. It is a common misconception that birth-derived tissue products, such as dual-layered dehydrated amnion–amnion graft (dHAAM), are similar regardless of the manufacturing steps. A commercial dHAAM product, Axolotl Biologix DualGraft™, was assessed for biological and mechanical characteristics. Testing of dHAAM included antimicrobial, cellular biocompatibility, proteomics analysis, suture strength, and tensile, shear, and compressive modulus testing. Results demonstrated that the membrane can be a scaffold for fibroblast growth (cellular biocompatibility), containing an average total of 7678 unique proteins, 82,296 peptides, and 96,808 peptide ion variants that may be antimicrobial. Suture strength results showed an average pull force of 0.2 N per dHAAM sample (equating to a pull strength of 8.5 MPa). Tensile modulus data revealed variation, with wet samples showing 5× lower stiffness than dry samples. The compressive modulus and shear modulus displayed differences between donors (lots). This study emphasizes the need for standardized processing protocols to ensure consistency across dHAAM products and future research to explore comparative analysis with other amniotic membrane products. These findings provide baseline data supporting the potential of amniotic membranes in clinical applications.

## 1. Introduction

### 1.1. Background: History and Clinical Utility of Membrane

The human amniotic membrane (hAM) is often considered surgical waste, devoid of ethical concerns, yet it serves as a highly accessible biomaterial with a vast array of applications in tissue repair and regenerative medicine. Sourced ethically from human birth tissues, particularly the placenta, hAM is rich in bioactive molecules, growth factors, cytokines, and extracellular matrix components that contribute to its regenerative and immunomodulatory characteristics [1]. Its key properties include a low risk of immunogenicity, stimulation of re-epithelialization, pain relief, and antimicrobial and anti-fibrotic effects. The mechanical attributes of hAM including permeability, elasticity, flexibility, and barrier functions are closely linked to the extracellular matrix proteins derived from its source tissues [2].

The clinical application of amniotic membranes dates back over a century, beginning in 1910 with skin transplants and burn wound dressings [3,4]. Advances in processing and storage techniques have increased the availability of hAM, allowing its application across various medical specialties, including ophthalmology, periodontology, plastic surgery, and chronic wound care [5,6,7,8]. The unique qualities of hAM, which facilitate cell migration, promote healing, and deliver a range of growth factors and proteins, have confirmed its clinical efficacy as an allograft. However, using human fetal membrane graft products poses challenges, including donor variability and the absence of standardized processing protocols. Various methods, such as decellularization, lyophilization, cryopreservation, and chemical sterilization, aim to preserve the structural integrity and biological properties of birth tissue allografts. The choice of processing method ultimately influences the membrane’s material properties, which must be balanced with biocompatibility, bioactivity, and mechanical strength for optimal clinical outcomes.

Recent studies focusing on Axolotl Biologix Graft and DualGraft™ products have contributed significantly to understanding their characteristics. While the existing literature and clinical applications provide insights into these products’ mechanisms, in-house and outsourced third-party studies have further examined the biocompatibility and mechanical properties of the DualGraft membrane product (dhAMM). Figure 1 illustrates the Axolotl DualGraft™ (dHAAM).

### 1.2. Biocompatibility

The hAM consists of an epithelial layer, a basement membrane, and a stromal layer, each contributing to its biological and mechanical properties that make it suitable for regenerative medicine. The epithelial layer, made up of simple cuboidal amniotic epithelial cells (AECs), features apical microvilli that facilitate solute and water exchange [9]. These cells also produce essential growth factors, including epidermal growth factor (EGF) and vascular endothelial growth factor (VEGF) [1]. AECs are recognized as multipotent, expressing stem-cell-specific transcription factors such as octamer-binding protein 4 and NANOG [10,11]. The collagenous basement membrane provides necessary support for fetal development and contains critical growth factors for fetal survival.

The stromal layer can be subdivided into three layers: the compact layer, the fibroblast layer, and the spongy intermediate layer. The compact layer, adjacent to the basement membrane, is composed of collagen types I and III and fibronectin. The fibroblast layer contains collagen types I, III, and VI, as well as various growth factors and cytokines that enhance the mechanical integrity of the amnion [12,13]. The spongy layer, loosely connected to the chorion, provides a protective cushion for the fetus and facilitates the amniotic sac’s fluid-filled environment [12,14,15]. The extracellular matrix (ECM) is integral to the amnion, providing structural support and enabling fetal expansion during pregnancy. Its unique composition bolsters regenerative capabilities, making hAM a valuable resource for clinical applications, particularly in wound healing and tissue regeneration [16,17]. Figure 2 illustrates the basic anatomical structure of the amniotic membrane.

Bacterial colonization significantly hampers wound healing, particularly due to the ESKAPE pathogens—*Enterococcus faecium*, *Staphylococcus aureus*, *Klebsiella pneumoniae*, *Acinetobacter baumannii*, *Pseudomonas aeruginosa*, and *Enterobacter* species—resistant to conventional antibiotics [18]. Chronic wounds have an increased infection risk due to prolonged exposure, emphasizing the need for effective barriers. The dense ECM network of hAM acts as a physical barrier against microbial invasion and protects underlying tissues, thus enhancing healing [18,19].

Beyond its physical barrier properties, hAM exhibits a range of antimicrobial properties that further its clinical efficacy. Antimicrobial peptides (AMPs) present throughout the amnion are critical to the innate immune response, protecting the fetus from infections. These AMPs can disrupt microbial cell membranes and inhibit replication, effectively reducing the activity of common pathogens found in chronic wounds [18,19]. By modulating immune cell activity, hAM fosters an environment conducive to healing. Various processing techniques, including gamma irradiation and antimicrobial coatings, can further enhance these properties while preserving the membrane’s integrity [19,20].

Amniotic epithelial cells (AECs) possess immunosuppressive capabilities, producing soluble factors that inhibit T-cell proliferation and inflammatory cytokine release [21,22]. This immunomodulatory effect, coupled with the presence of molecules like HLA-G, aids in promoting a favorable healing environment and enhances tissue acceptance in clinical scenarios [21,22,23,24,25,26]. Understanding the impact of residual factors in dHAAM on therapeutic potential will be key in future research endeavors.

### 1.3. Mechanical Properties

Mechanical properties are essential for understanding the performance and utility of biomaterials like the human amniotic membrane (hAM). These properties, including tensile strength, elasticity, and flexibility, determine how well the membrane can withstand physical stresses and maintain its structural integrity during clinical application. In regenerative medicine, hAM’s mechanical characteristics provide a stable structural scaffold that supports tissue repair and regeneration. The ability of hAM to retain these properties, even after processes like dehydration and decellularization, can significantly influence its effectiveness in various clinical settings, assessing these factors essential for optimal therapeutic outcomes. The design of a simple biomaterial, such as dHAAM, necessitates both a synergistic biocompatibility component along with an integration of mechanical properties. The mechanical properties tested within this characterization product review include the ultimate suture strength, tensile strength, and compressive and shear modulus assessments. Establishing the mechanical properties of a biomaterial based on linear and elastic relationships between loads acting or reacting with the material aids in determining the critical points of stress. These properties are highly essential for the fabrication of tissues such as skin and cartilage because these tissues possess the capacity to support cell proliferation and extracellular matrix deposition. For repairing and regenerating tissue, the biomaterial must provide sufficient mechanical support to endure in vivo stresses and load-bearing cycles.

Suture strength is a critical factor to consider for biomaterials due to the efficacy, safety, and effectiveness of treatment. Suture strength directly impacts the ability of an allograft to hold tissues together by acting as a structural barrier that facilitates effective healing and infection control and reduces risk of wound dehiscence. With a high suture strength, a material can withstand mechanical stresses and reduce the need for excessive suturing, which can minimize infection potential and inflammation in the surrounding tissue. It affects both the immediate outcomes of surgical repairs and the long-term success of biomaterial implants and other devices.

Tensile strength refers to the maximum stress that a material can withstand before it ruptures or breaks under tension. According to Veeman et al. [27], the tensile strength of AM can vary depending on factors such as tissue source, donor characteristics, processing techniques, and preservation methods. Tear resistance is the ability of a material to resist the propagation of a tear or crack when subjected to mechanical forces. Amniotic membranes exhibit considerable tear resistance due to their fibrous structure and interwoven collagen fibers, adding to their importance for surgical applications where AM is used as a graft or patch for repairing damaged tissues or covering wounds [28]. Both tensile strength and tear resistance are useful to consider the suture strength of AM.

The elastic modulus, also known as Young’s modulus, measures the stiffness or rigidity of a material. It represents the slope of the stress–strain curve in the elastic deformation region. A decellularized amniotic membrane typically has a relatively low elastic modulus compared to synthetic materials or native tissues, which allows it to conform to irregular surfaces and withstand mechanical deformations without excessive stress concentration [1,28]. The current literature demonstrates that during uniaxial testing, hydrated (wet) dog-bone-shaped amniotic membranes have a 4–5 MPa range [28]. This results in great flexibility and conformability, allowing them to adapt to the shape and contours of various anatomical surfaces. This property is advantageous for applications such as ocular surface reconstruction, where AM is used to repair corneal defects or treat ocular surface disorders. Viscoelasticity refers to the time-dependent response of a material to mechanical loading, characterized by both elastic (reversible) and viscous (irreversible) behavior. Amniotic membranes exhibit viscoelastic properties, with stress relaxation and creep occurring over time when subjected to constant or cyclic loading. This viscoelastic behavior is important for maintaining tissue integrity and function under physiological conditions. The thickness and porosity of an amniotic membrane can influence its mechanical properties, including tensile strength, elasticity, and permeability to fluids and nutrients. Thicker membranes may have higher tensile strength but lower flexibility, while increased porosity can enhance cellular infiltration and tissue integration but may compromise mechanical integrity.

This comprehensive analysis of the Axolotl Biologix DualGraft™ membrane product (dhAMM) provides a detailed understanding of its mechanical performance, ensuring its suitability for various clinical uses. Through evaluating these critical properties, the study aims to enhance the application of hAM in regenerative medicine.

## 2. Materials and Methods

### 2.1. Axolotl Biologix dHAAM Processing Method

The Axolotl Biologix dehydrated dual-layer amniotic membranes (dHAAMs) are prepared from amnion–chorion tissue acquired from consenting donors undergoing either cesarean section or vaginal birth. Donors are screened following a rigorous tissue acceptance program and tested for communicable diseases such as Hepatitis B and C, HIV 1/2 HTLV I/II, Syphilis, and other pathogenic microbes, adhering to 21 CFR 1271 and American Association of Tissue Banks (AATB) standards. Processing begins with the separation of amnion from the chorion membrane using blunt dissection, which is then washed and cleaned of residual blood and tissue debris. To create the dual layering, the membrane is folded stroma side to stroma side, with the amnion epithelial side facing out. The membrane is then dehydrated, cut to size, pouched, and sealed in the final product configuration before being gamma-irradiated to a sterility assurance level (SAL) of 10^−6^ per ISO 11137-2 [29].

### 2.2. Biocompatibility Testing Included the Following Assays

The biocompatibility testing included an in-house cellular viability assay with luciferase-based ATP detection, a visual of cell adherence with Cell Tracker Green CMFDA on confocal microscopy, an antimicrobial disk diffusion study, and a third-party outsourced full proteomic characterization.

### 2.3. Antimicrobial

The antimicrobial properties of dHAAM were tested against three common microbial species known to cause nosocomial infections utilizing the Kirby–Bauer disk diffusion susceptibility testing protocol for bacteria and yeasts [30,31]. All strains were identified as pure and DNA-verified and were acquired through ATCC^®^: *Candida albicans* (Robin) Berkhout ATCC^®^ 18804, *Escherichia coli* ATCC^®^ 25922, and *Staphylococcus epidermidis* (Winslow and Winslow) Evans ATCC^®^ 12228. However, instead of using the MacFarland method for obtaining the turbidity as an analog for cfu (colony forming units) for each organism, we utilized the Shimidzu UV-mini 1240 spectrophotometer. The study design included 3 different dHAAM product lots, each with eight 6 mm circles hand-cut in a sterile environment using a 6 mm sterile biopsy punch.

This approach helped us to evaluate the consistency and effectiveness of the antimicrobial properties across different tissues, ensuring a comprehensive assessment of the product’s performance. The following antimicrobial disks were selected to match the susceptibility level of each organism: Gentamicin 10 microgram (mcg), Chloramphenicol 30 mcg, Tetracycline 30 mcg, and Fluconazole 12.5 mcg. Since the fluconazole disks are not available commercially, a 2 mg/mL solution was made and 12.5 uL of the solution was placed onto the disks and dried at 50 °C for 24 h [32].

The lyophilized microbes were reconstituted according to supplier guidelines with an initial suspension in broth media and subsequently cultured on agarose plates. Media for initial culturing were made onsite and were specific to each organism. Dilutions of *E. coli*, *S. epidermidis*, and *C. albicans* suspensions were created in concentrations of 1.15 × 10^8^ cfu, 1.18 × 10^8^ cfu, and 2.05 × 10^6^ cfu, respectively, in a 0.85% sterile saline solution. The Mueller Hinton (MH) agarose plates were then inoculated using sterile swabs and allowed to sit for 3–5 min prior to application of the treatments. For *C. albicans*, the MH agar plates contained 2% glucose per the NCCLS guidelines. Two antibiotics disks, one control disk, one membrane, and one amniotic-derived cell-conditioned media-treated disk was applied to each plate, except for the *C. albicans* plates where only one antifungal, fluconazole, was added instead of an antibiotic. Plates with bacteria were placed into an incubator at 37 °C for 16 h, whereas the yeast was incubated at 35 °C for 24 h until observation and the documentation of the final results.

### 2.4. Cell Compatibility/Viability

To assess the cellular viability and compatibility of dHAAM, cultured adult human dermal fibroblasts were seeded onto membrane samples and visualized using scanning electron microscopy (SEM) and confocal microscopy imaging techniques. The cells used for the subsequent experiments were cultured as follows: A vial of adult human dermal fibroblasts (hDFs), passage 2, derived from normal human facial dermis (provided by Cell Applications Inc., San Dieog, CA, USA) was thawed and cultured in Dulbecco’s modified Eagle’s medium 1X (DMEM) containing 10% fetal bovine serum (FBS) and incubated in a 5% CO_2_ incubator at 37 °C. The culture medium was replaced 24 h after flask inoculation then replaced every other day with fresh complete DMEM. When cells’ confluency reached 70–80%, cells were subcultured using TrypLE Select solution (Gibco, Grand Island, NY, USA), and a single cell suspension was created and then counted using a hemocytometer and Tyrpan Blue.

To begin SEM setup, four 10 mm biopsy samples of the membrane were prepared in a 48-well Corning plate, each well filled with 1 mL of DMEM complete media and incubated overnight for acclimation. A sterilized metal ring weight was placed on each membrane to prevent flotation. Subsequently, cultured passage-3 human dermal fibroblasts were seeded onto each membrane at 10,000 cells/cm^2^, with the plate area being 0.95 cm^2^. The plate was incubated in 5% CO_2_ for 24 and 48 h to allow for cell attachment and doubling. After the incubation period, samples were fixed with 2.5% glutaraldehyde in sodium cacodylate. For SEM analysis, transverse images of the membranes, both with and without cells, were taken. These included fiber measurements and were processed at the Northern Arizona University (NAU) Imaging & Histology Core Facility (IHCF), where samples were sputter-coated and imaged at magnifications ranging from 129× to 3500×.

For further visibility, CellTracker^TM^ Green CMFDA fluorescent dye (from ThermoFisher/Invitrogen, Waltham, MA, USA) was utilized to monitor live versus dead adult hDF cells’ occurrence and proliferation when cultured with dHAAM membrane. Following the necessary fluorescent dye concentration optimization (2.5 µM in serum-free DMEM) and hDF cell culture process, cells were seeded onto 4 mm dHAAM biopsy samples (density 10,000 cells/cm^2^) and incubated for 72 h in 5% CO_2_ incubator at 37 °C. Cell dye dilution was crafted following the product experimental protocol using DMEM. After incubation, 2.5 µM dye solution was added to each sample and incbuate for an additional 30 min before samples were fixed with 4% PFA in PBS and then mounted on clean and labeled glass slides using ProLong Glass Antifade Mountant (from ThermoFisher). Control samples of dHAAM without cells were also processed using a similar technique. Samples were viewed and images were collected via confocal microscopy (Leica TCS SPE II) with a 488 nm laser.

To further determine the number of viable cultured cells on membranes, we utilized the Cell Titer-Glo^®^ Luminescent Cell Viability assay (from Promega, Madison, WI, USA), which quantifies ATP presence, expressing metabolically active cells. The assay relies on the properties of a proprietary thermostable luciferase generating a stable “glow-type” luminescent signal with a half-life from this reaction being greater than five hours. The basic assay design included hDF cultured in the presence or absence of DualGraft membrane, with quadruplicates of the following in a 96-well cell culture plate: a media control (DMEM), a cell control (cultured hDF from Cell Applications, San Diego, CA, USA) with a 6000 cells/cm^2^ seeding density, dHAAM control (no cells, 4 mm biopsy punches in medium), and dHAAM with cells (6000 ells/cm^2^ seeding density onto 4 mm biopsies). Three different lots of dHAAM were tested for comparison. The membrane biopsies were hydrated overnight with complete DMEM before subcultured cells were seeded into the corresponding plate wells, while the rest of the samples were prepared with fresh complete DMEM before the test plate was incubated for 72 h at 37 °C in a 5% CO_2_ incubator. The Cell Titer-Glo reagent was prepared; the test plate was equilibrated to room temperature with 100 µL of media removed from each well before directly adding 100 µL of Cell Titer-Glo reagent to each well following the assay protocol; we read the luminescence with the gain set to 135 on a BioTek Cytation 1 image reader. Luminescence was recorded and data analysis comparison was completed.

The final biocompatibility test conducted on dHAAM was a full proteomic characterization identifying the extracellular proteins, growth factors, and cytokines present. We provided three dHAAM samples of different lot numbers to Biognosys AG (Next Generation Proteomics) to determine a global proteome. The samples were shipped and stored at ambient temperatures and then prepared for mass spectrometry (MS) using Biognosys’ optimized protocol comprising an albumin depletion of biofluids (Pierce Albumin Depletion Kit by Thermo Scientific) and re-solubilization of membrane samples in 200 µL of Biognosys lysis buffer using a Precellys evolution homogenizer (Bertin Technologies). Samples were further prepared according to Biognosys SOP, which includes reduction, alkylation, and digestion to peptides using LysC (Wako Chemical, 1:200 protease to total protein ratio) and trypsin (Promega, 1:50 protease to total protein ratio) per sample overnight at 37 °C. Peptides were desalted using an OasisHLB 96-well 2 mg sorbent plate (Waters) according to the manufacturer’s instructions and dried down using a SpeedVac system. HRM data analysis was completed for both sample types. Then, peptides were resuspended in 1% acetonitrile and 0.1% formic acid (FA) and spiked with Biognosys iRT kit calibration peptides. Peptide concentrations were determined using an mBCA assay (Pierce, Thermo Fisher). For Data Independent Acquisition (DIA) liquid chromatography–mass spectrometry (LC-MS) measurements, 3.5 µG of peptides was injected into an in-house packed reversed phase column on a Thermo Fisher Scientific^TM^ EASY-nLC 1200 nano-liquid chromatography system connected to a Thermo Fisher Scientific^TM^ Orbitrap^TM^ Exploris 480 mass spectrometer equipped with a Nanospray Flex^TM^ ion source and an FAIMS Pro ion mobility device (Thermo Fisher Scientific). The liquid chromatography (LC) solvents were water with 0.1% FA (solvent A) and 80% acetonitrile with 0.1% FA in water (solvent B). The nonlinear LC gradient was 1–50% solvent B in 210 min followed by a column washing step in 90% B for 10 min, and a final equilibration step of 1%B for 8 min at 60 °C with a flow rate set to 250 nL/min. The FAIMS-DIA method consists of an applied compensation voltage of one full-range MS1 scan and 34 DIA segments as adopted from Bruderer et al. (2017) and Tognetti et al. (2022) [33,34].

Using patented Hyper Reaction Monitoring (HRM) technology, MS data are acquired in DIA mode in a highly parallel process, providing a comprehensive peptide-level measurement of the detected proteins from the sampled dHAAM, generating a comprehensive spectral library (HRM map) that is further analyzed using quality control (QC) metrics for identifications. Minimal normalization was performed across all runs, indicating a reproducible sample preparation, constant peptide amount, and equal sensitivity of all instruments across all runs. All solvents were HPLC-grade from Sigma-Aldrich (St. Louis, MO, USA).

### 2.5. Suture Strength

Ultimate suture strength (USS) quantifies a material’s ability to resist tearing after being sutured with an interrupted suture from the hydrated graft material during tension (pulling). USS was tested with an HR-2 rheometer (TA Instruments, New Castle, DE, USA) with dHAAM samples wetted with PBS and compared at room temperature (21 °C) for a total of two trials (n = 2). Four rectangular dHAAM samples (L × W × H: ~13 mm × 50 μm × 20 mm), from each study article lot, underwent suture strength testing with the rheometer, using a 25 mm × 5 mm tensile clamp apparatus. Two rectangular samples from the same lot were stacked on each other and hydrated. A 2-0 prolene suture was placed 3 mm from the top center of the two samples, using an interrupted surgical tie. The loose end of the suture was clamped in the top fixture, and 5 mm of sample bottom height was clamped in the bottom fixture. For suture strength, the upper clamp with the suture was pulled apart vertically at a rate of 50 μm/s. The rheometer reported pull force (N) across the distance of suture pull. USS for the 2-0 suture was determined as maximum pull force (N) divided by the area (mm^2^) of the suture in contact with the dHAAM (50 μm × πr, (r = radius of the suture (0.15 mm), area = 0.024 mm^2^).

### 2.6. Tensile Modulus

The tensile modulus of a material is defined as its ability to resist deformation when a tensile (pulling) force is applied to the material. However, biologic materials often exhibit a viscoelastic response to stress as there is often a time-dependent deformation; for example, after pinching the skin of the knuckle, the skin takes time to relax back to its original shape after the initial deformation from pinching. This is caused by a viscous component in the material that causes energy in the material to dissipate as heat. To account for the viscoelasticity of the material, the complex tensile modulus (E) was determined. The complex tensile modulus quantifies the ratio of tensile stress to tensile strain (stiffness) of the graft material and differentiates the viscous (loss of energy) and elastic (storage of energy) moduli of the sample, which determine the viscoelasticity of the graft material in tension. A total of two samples per dHAAM lot were tested to evaluate both dry and hydrated (with phosphate-buffered saline (PBS)) conditions at room temperature (21 °C), assessing three different tensile forces (0.1 N, 0.225 N, and 0.35 N). Samples were cut into two rectangular pieces (13 mm × 50 μm × 20 mm) from each lot and underwent tensile testing with a DHR-2 hybrid dynamic mechanical analysis (DMA) rheometer (TA Instruments, New Castle, DE, USA), using a 25 mm × 5 mm tensile clamp apparatus. Samples were clamped at the top and bottom 5 mm of their height, leaving an initial unclamped length (l_o_) of ~10 mm. Prior to mounting, the hydrated samples were saturated with PBS, pH 7.4, for 1 min before mounting. Dry samples were first tested by slowly raising the rheometer head and applying tensile stress until the measured axial force reached 0.1 N to begin. At 0.1 N, oscillation was set to 20 µm. The rate of oscillation (applied and released tension on the sample) began at 1 rad/s and was increased, sweeping across a range up to 20 rad/s. This non-destructive test was completed three times at the same starting force (0.1 N) for the same conditioned sample. Three additional trials were completed for the same sample with a starting tensile force of 0.225 N and oscillation of 25 µm, with the final three trials repeated at a starting force of 0.3 N and oscillation of 30 µm. The wet samples’ hydration was maintained between each test by wetting both sides with PBS. These tests were all completed with an additional sample from the same lot, and then two additional samples from the second and third lot (6 samples in total). Due to the low strain and forces applied to the material, the test was non-destructive and the same dHAAM samples were able to be used for further ultimate tensile strength testing (destructive) [35]. The average tensile modulus and standard deviation were calculated for each studied sample.

### 2.7. Tensile Strength

Ultimate tensile strength (UTS) quantifies the strength required to tear the graft material during tensile (pulling) force. The tensile strength testing involved both dry and hydrated dHAAM samples. Samples were again tested as either dry or hydrated with a total of two trials of each dHAAM lot (3 total), completed at 21 °C. Two rectangular dual-layer graft samples (L × W × H: 12.5 mm × 50 µm × 20 mm) from each study article lot (n = 2) underwent tensile testing with the DMA rheometer, using the same 25 mm × 5 mm tensile clamp apparatus, leaving an initial unclamped length of (l_o_) of ~10 mm. Mounted tensile samples were hydrated with PBS, pH 7.4, for 1 min prior to wet sample testing. For tensile strength assessment, the upper clamp was pulled vertically at a rate of 50 μm/s. The DHR-2 hybrid rheometer reported tensile strength (Pa) as pull force (N) divided by cross-sectional area (0.65 mm^2^). The test was performed and repeated two times for each sample lot. The average UTS and E, along with standard deviations, were calculated for each sample and presented in bar graphs.

### 2.8. Shear Modulus

With a biological material, the structure of the material is not always uniform in all directions. Shear forces are applied perpendicularly to tensile or compressive forces. Complex shear modulus (G*) quantifies the resistance (stiffness) of the graft material to shearing apart (delamination) and differentiates the viscous (loss) and elastic (storage) moduli of the sample, which can determine the viscoelasticity of the sample in the shear plane of action. The complex shear modulus was tested with a DHR-2 hybrid rheometer across a physiologically relevant oscillation rate range of 1 rad/s, up to 20 rad/s (0.16–3.2 Hz, 10–192 BPM). In total, three dHAAM tissue donor lots were tested with three trials for each study sample completed at body temperature (37 °C). The dHAAM samples were cut into 20 mm circles, and 6 circles were stacked to increase sample height. The samples were placed on a temperature-controlled Peltier plate and hydrated with PBS, pH 7.4, for 1 min prior to testing. Sample thickness increased ~33% (from 50 μm dry to 67 μm hydrated). Samples were then compressed with a 20 mm plate head by 25–30% of the initial height (0.1–0.3 N of compression) to begin the shear modulus testing. Samples underwent 1% shear, across a physiologically relevant shear rate (1 rad/s, up to 20 rad/s) at body temperature (37 ± 0.5 °C). This non-destructive test was repeated 3 times (n = 3). Hydration was maintained between each test by wetting the edges of the sample on the Peltier plate with PBS. Since the test was non-destructive, the samples were used for further compressive modulus testing [35]. The rheometer reported the modulus as the change in stress over the change in strain (slope) with respect to frequency; for each sample, three repeat modulus measurements were conducted.

### 2.9. Compression Modulus

Complex compression modulus (E*) quantifies the ratio of stress to strain (stiffness) of the graft material and differentiates the viscous (loss) and elastic (storage) moduli of the sample, which can determine the viscoelasticity of the sample in compression. Samples from the previous shear modulus testing were then tested by maintaining 25–30% of the initial height (0.1–0.3 N of compression). A steady state of compression (0.1 N–0.3 M) at 25–30% was reached before the dynamic compressive stress (up to 3% of starting height) was applied and released across the linear frequency sweep from 1.0 to 20.0 rad/s. A 3% dynamic compressive dynamic stress provides more consistent data and was verified by comparing the ratio of compressive modulus to shear modulus, which for isotropic material is 3:1 and for most biological materials does not exceed 10:1 [35]. The rheometer reported the modulus as the change in stress over the change in strain (slope) with respect to frequency. For each sample, three repeat modulus measurements were recorded (n = 3). Hydration was maintained between each test by wetting the edges of the sample on the Peltier plate with PBS.

## 3. Results

### 3.1. Antimicrobial Results

All test plates were visually checked at the end of the incubation period to confirm organism colony characteristics and to ensure there was no contamination within the cultured plates. All test plates showed no inhibition (0 mm) of growth for all dHAAM membranes tested (See Figure 3). The zones of inhibition for all antibiotics were in the expected susceptible range for each microorganism (see Table 1 below): Chloramphenicol: >18 mm; Gentamicin: >15 mm; Tetracycline: >15 mm; and Fluconazole: >19 mm [29]. Two test plates for each organism (six total) showed no growth, confirming there was no contamination.

It is possible that the diffusion-based test can result in a false negative because the dried membrane products may lack diffusible antimicrobial properties. It is known in the literature that allograft membranes and amniotic fluid derivatives contain inherent antimicrobial properties [7,36,37]. Based on the results, these properties for dHAAM do not diffuse into agarose and did not provide a zone of inhibition. The results of this initial study suggest that additional research is needed to understand the antimicrobial properties of dHAAM, as the literature does support the antimicrobial capabilities, growth factor, and peptides found in the amniotic membrane [38].

### 3.2. SEM Image Results

Thin layers of cellular tissue cover the exposed membrane surface, with latent constructs resembling single cells stretching across the landscape. The 24 h and 48 h cell incubation samples resembled the same topography of cellular structures on the control membrane samples (see Figure 4). With dHAAM being non-decellularized (during its manufacturing processing), it was difficult to see whether the observed cell structures were from newly attaching cells or part of the membrane; see the images below. To address this issue of cellular compatibility visualization, we refer to the CellTracker Green assay for clarity and the presence of new cellular structures and cellular biocompatibility.

### 3.3. CellTracker Green Fluorescent Probe

CellTracker fluorescent probes are dyes used to monitor cell movement, location, proliferation, migration, and invasion. They have been designed to freely pass through cell membranes and, once inside, are transformed into cell-impermeant reaction products. After conversion, the fluorescent probes are retained in living cells through several generations and can display fluorescence for at least 72 h and exhibit ideal tracking dye properties and display no cytotoxicity. The samples of dHAAM membrane seeded with adult hDF successfully displayed cell presence and attachment (see the following images (Figure 5)), while control samples showed no signs of dye present—supporting the hypothesis that dHAAM encourages the attachment, growth, and proliferation of adult hDF, indicating the biocompatibility of dHAAM.

### 3.4. Cell Titer-Glo Assay Results

The CellTiter-Glo^®^ Luminescent Cell Viability Assay (from Promega) is a homogeneous method for determining the number of viable cells in culture based on the quantitation of ATP. The procedure involves adding a single reagent (CellTiter-Glo^®^ Reagent) directly to cultured cells in a serum-supplemented medium. The addition of reagent results in cell lysis and the generation of a luminescent signal proportional to the amount of ATP present, which is proportional to the number of viable cells present. By comparing the metabolic activity of cells in either the presence or absence of dHAAM, a significant difference is present (*p* = 0.00021) (Anova test); see Figure 6. Each dHAAM lot presented a higher luminescence than adult hDF cells alone. These results indicate a higher population of cells present when cultured with dHAAM samples.

### 3.5. Proteomics Results

The mass spectrometric data were analyzed using the Biognosys search engine SpectroMine (version 3), with the false discovery rate of peptide and protein levels set to 1%. A human UniProt fasta database (*Homo sapiens*, 1 July 2022) was used for the search engine, allowing for two missed cleavages and variable modifications (N-term acetylation, methionine oxidation). A proteome-wide protein profile was generated using HRM-Quality control, with numbers of identified proteins, peptides, and peptide ion variants in each sample (see Figure 7) for an average total of 7678 unique proteins, 82,296 peptides, and 96,808 peptide ion variants. The HRM mass spectrometric data were analyzed using Spectronaut Pulsar software (Biognosys, version 16), and an assay library was generated.

To further dissect the influx of data and numerous proteins and peptides identified from Biognosys analysis, the data were reformatted and input into the matrisome analysis program, Matrisome AnalyzeR. The primary aim was to further organize the identified intensities from the three unique dHAAM lots tested [39]. The compendium of all genes encoding ECM and ECM-associated proteins is termed the ‘matrisome’ and classifies components into different structural or functional categories such as matrisome-associated, core collagen, core proteoglycans, etc. This nomenclature is largely used to annotate ‘-omics’ datasets and help researchers classify and tabulate large molecule datasets [38]. After inputting the data, a bar graph (matribar) representing the total number of matrisome molecules classified according to core matrisome, matrisome-associated and non-matrisome proteins within the dataset is generated (Figure 8). Upon further breakdown of the matrisome categories across the entire dHAAM dataset, a pie chart was crafted (Figure 9) to further classify the core matrisome (i.e., genes encoding structural components of the ECM—collagens, proteoglycans and ECM glycoproteins) and matrisome-associated (i.e., genes encoding non-structural components of the ECM—regulators, secreted factors and other ECM-affiliated proteins); see Figure 8 for the annotated matrisome divisions and Figure 9 for the matrisome categories [39].

### 3.6. Mechanical Testing Results

#### 3.6.1. Suture Strength Results

Two repeats (n = 2) of two stacked and hydrated samples were tested from lots 1170, 1231, and 1454 of the dHAAM. Results showed a trend towards increased USS for lot 1170 over lots 1231 and 1454; see Figure 10. However, all datasets were statistically equivalent in average suture pull force amongst the three donors (*p* value = 0.19 (1170 vs. 1231), 0.89 (1231 vs. 1454), and 0.35 (1170 vs. 1454)), with an average pull force of ~0.40 N per two dHAAM samples (0.2 N per sample). Considering the 2-0 prolene suture diameter, this pulls force corresponds to a pull strength of 8.5 MPa for the dHAAM samples.

The dHAAM samples have an average suture pull force of 0.2 N when a 2-0 prolene suture is pulled from the samples. This corresponds to a dHAAM pull strength of 8.5 MPa. Assuming the pull strength is constant, the full force needed to remove sutures of various sizes can be extrapolated. As seen in Table 2 below, the pull force ranges from 0.05 N for a 6-0 suture to 0.23 N for a 0-sized suture.

#### 3.6.2. Tensile Modulus Results

For each dHAAM lot studied, two rectangular 12.5 mm × 50 µm sample sizes offered the largest cross-sectional area for improve the resolution of the rheometer mechanical testing datasets. Scatter plots of tensile modulus (E) versus rate of applied tension (rad/s) were created for all three scenarios with the average tensile modulus and standard deviation (error bars) included. The average tensile modulus and standard deviation (error bars) for the trials of each dHAAM scenario contain each tension rate, along with a scatter plot of the increase in modulus with force applied.

For the dry dHAAM lot samples’ analysis, a total of four rectangular samples were tested for the complex tensile modulus; two samples from each lot were repeat tested three times with a starting tensile force of 0.1 N, 0.225 N, or 0.35 N. The dry samples’ complex tensile moduli data were approximately 110 MPa and 115 MPa at 0.1 N, 185 MPa and 200 MPa at 0.225 N, and 240 MPa and 255 MPa at 0.35 N; see Figure 11 below. There was a linear increase in tensile modulus with an increase in applied tensile force for the two dHAAM lots tested. Although the data trends similarly between lots, there was a significant difference (*p* < 0.01) in complex tensile modulus between the lots at each applied axial force; however, the increase in modulus was nearly linear with respect to the applied force (R^2^ > 0.98).

For the wet sample analysis, a total of six rectangular samples were tested for complex tensile modulus; two (n = 2) samples from each of the three lots (n = 3) were repeat tested three times (n = 3) at a starting tensile force of 0.1 N that increased to 0.225 N and 0.3 N. A linear increase in tensile modulus with an increase in applied tensile force was observed for all lots (R^2^ > 0.99—see Figure 12). There was a significant difference in complex tensile modulus between all lots, with lot 1454 having the lowest tensile modulus of all samples, whereas lot 1170 (E = 1.8) was ~1.9× higher, and lot 1231 (E = 2.7) was ~3.1× higher than lot 1454. The tensile modulus of the wet samples indicates that tensile modulus trends similarly among different donors of dHAAM, although the moduli themselves may vary.

Overall, the complex tensile moduli for hydrated samples were 5× lower than the dry samples at all axial forces, indicating a greater elasticity and reduced stiffness when dHAAM samples are wet. There is linear increase in tensile modulus with an increase in applied tensile force for all the wet sample lots (R^2^ > 0.99, see Figure 13). The tensile modulus data indicated a similar trend among different donor lots, although the moduli themselves may vary. Tensile modulus decreases when the membrane is wet, meaning it can be manipulated (pulled, compacted, stretched) into a wound bed without tearing.

#### 3.6.3. Tensile Strength Results

Similarly to the dynamic tensile modulus results, the tensile strength varied significantly between donors. For each lot, as the UTS increased, so did the modulus. Two (n = 2) samples from each of the three lots (n = 3) were tested. Donors 1170 and 1454 had significantly lower tensile strength and modulus than donor 1231. Lot 1454 had the lowest tensile strength of all samples. Lot 1170 had a tensile strength ~2× higher than lot 1454. Lot 1231 had a tensile strength ~8.5× higher than lot 1454. Two sets of data analysis were completed, leading to a scatter plot of tensile strength versus strain (Figure 14) and bar graphs (see Figure 15) of the average modulus (E) and tensile strength (UTS) at 21 °C.

#### 3.6.4. Shear Modulus Results

In total, eighteen circular samples were tested for complex shear modulus; six samples from each lot were repeat tested three times and compared at a 6 rad/s (1 Hz) strain rate. Lot 1454 exhibited a complex shear modulus (G*) of ~5.1 KPa (0.005 MPa) at 37 °C. Lot 1231 exhibited a G* of ~5.8 KPa (0.0058 MPa), although not statistically similar to lot 1454 (*p* value < 0.01); the difference between the two lots was <13%. Lot 1170 exhibited a G* of ~4.0 KPa, significantly different from lots 1454 and 1231 (*p* value < 0.01), with a 24% and 37% difference, respectively. Two sets of data analysis were completed: scatter plots of shear modulus (G*) versus shear rate (rad/s) (see Figure 16) with the average G* and standard deviation (error bars) for the three trials of each scenario included, as well as bar graphs of the average shear modulus at a physiologically relevant shear rate (6 rad/s = 1 Hz), were created for all three dHAAM studied. Assessing the shear modulus data confirmed that all three donor lots of dHAAM were statistically different, but within 45% of each other.

#### 3.6.5. Compressive Modulus Results

In total, 18 circular samples were tested for their complex compression modulus, with six samples from each of the three dHAAM lots repeat tested three times. Lot 1454 exhibited a complex compression modulus (E*) of ~30 KPa (0.03 MPa) at 37 °C. Lot 1231 exhibited an E* statistically similar to 1454 (E*~27 KPa, *p* = 0.157), with a 10.5% difference between lots. Lot 1170 exhibited an E* of ~37 KPa and was statistically different from lots 1454 and 1231 (*p* value < 0.01), with a 21% and 31% difference, respectively (see Figure 17).

Overall, the mechanical testing results revealed that the tensile modulus (stiffness) varied significantly between the dHAAM donor lots. For each dHAAM lot, as the tensile strength increased, so did the stiffness. While each dHAAM lot strength and modulus varied, the samples within each lot were similar. Assessing the shear modulus data confirmed that dHAAM donor tissues were statistically different, but within 45% of each other. Assessing the compressive modulus data confirmed that the compressive moduli of lots 1231 and 1454 were statistically similar (*p* = 0.157). Lot 1170 was statistically different (*p* < 0.01), but all three tested lots were within 31% of each other. Suture strength showed some variation between dHAAM lots; however, statistically, the variation was not deemed significant. The pull tear force was found to be ~0.2 N per sheet of hydrated dHAAM when suturing with a 2-0 suture.

## 4. Discussion

The findings of this study offer insights into the biological and mechanical characteristics of a commercially available dehydrated human amniotic membrane, Axolotl DualGraft™. dHAAM is one type of skin substitute product, often classified as Human Cells, Tissues, and Cellular and Tissue-Based Products (HCT/Ps). The U.S. Food and Drug Administration (FDA) regulates HCT/Ps under Title 21 of the Code of Federal Regulations (CFR) Part 1271. The regulations are designed to prevent the introduction, transmission, and spread of communicable diseases and to ensure the safety, purity, and potency of these products. Often, dHAAM based products are believed to be similar regardless of the processing practices of the birth tissue to synthesize the final finished biologic tissue product. In recent years, the research and clinical field has acknowledged that even minimal-manipulation processing techniques can lead to different end product characteristics, which could have varying clinical efficacy impacts [2,5,40,41].

In this study, a detailed biological and mechanical evaluation occurred. Analysis included antimicrobial, cellular biocompatibility, and proteomics analysis, as well as suture strength and tensile, shear, and compressive modulus testing. The prior literature reports the antimicrobial characteristics of amniotic membranes [42]. However, the lack of observed antimicrobial resistance in this study suggests that the dHAAM may have been impacted due to tissue processing techniques, including the dehydration step. Additionally, soluble peptides or growth factors, such as defensin, elafin, certain cytokines, and factors synthesized by the AECs and MSCs, could have been removed by way of the processing steps. Future research could focus on the identification and quantification of specific antimicrobial peptides, growth factors, and proteins prior to processing and then again post processing. Additional work could be achieved through techniques such as ELISA, where the membrane is enzymatically degraded to release the peptides for analysis.

The human amniotic membrane is biocompatible and has been repeatedly demonstrated to have a low immunogenicity [43,44,45,46,47,48]. Biocompatibility testing using the Cell Titer-Glo assay demonstrated increased metabolic activity in the presence of adenosine triphosphate (ATP) when dHAAM with cells was present. An additional biocompatibility test used a fluorescent dye to view human dermal fibroblasts that had been seeded on the dHAAM. The data demonstrated that dHAAM with hDFs demonstrated successful attachment and growth compared to the control. These data could indicate that the amniotic membrane aids in cellular proliferation via proliferative gene upregulation and/or cellular pathway signaling [49]. Additional previously published work also suggests that the use of cryoprotectants and simple freezing methods can impact cellular growth and proliferation [50]. Since the Axolotl DualGraft™ does not contain cryoprotectants and is freshly frozen during processing, these previously reported findings offer insight into the additional reasons why the dHAAM is highly biocompatible.

The last biological evaluation was a proteomics assessment. Using the annotated matrisome categories, the ten most prevalent core matrisome, ECM glycoproteins, collagens, proteoglycans, and ECM regulators were further evaluated; see Table 3, Table 4, Table 5, Table 6 and Table 7 below [51]. Each protein is briefly described, noting the proteins’ annotated gene, name, description, and function. Future characterization efforts could entail comparing varying processing techniques for the matrisome outputs. One item to note would be a way to control the biologic variability between maternal tissue donors, as the proteomic analysis could have variability in total protein content and matrisome intensity across different lots (based on the donor) of dHAAM. This variability highlights the inherent differences that can occur even with manufacturing processes. These results underscore the importance of establishing standardized criteria for the processing of amniotic tissues and donor tissue acceptance beyond communicable disease testing and donor risk assessment interviews (DRAIs).

Understanding the mechanical properties of amniotic membranes is equally important to that of the biologic characteristics. The mechanical properties have a direct impact on clinical efficacy and usability in health care providers’ practices. In this study, suture strength, tensile, shear, and compressive modulus testing occurred.

The tensile strength of dHAAM is a critical factor, referring to the maximum stress the material can withstand before rupture. This property can vary based on tissue source, donor characteristics, processing techniques, and preservation methods. For example, the tensile strength of fresh amniotic membrane can range from 0.1 to 1 MPa, while the tensile strength of dehydrated or dry amniotic membrane is higher at 1–2 MPa, and crosslinking the membrane with a crosslinking agent such as glutaraldehyde can increase the tensile strength to 2 MPa or more [52]. The fibrous structure of amniotic tissue with interwoven collagen fibers imparts significant tear resistance, making it a potentially suitable HCT/P for surgical applications where it can act as a structural graft, wrap, or patch.

The elastic modulus, or Young’s modulus, measures a material’s ability to resist deformation due to an applied compressive or tensile stress. The amniotic membrane typically exhibits a relatively low elastic modulus, allowing it to conform to irregular surfaces and endure mechanical deformations without excessive stress. This property is particularly advantageous for applications requiring flexibility and conformability, such as ocular surface reconstruction. However, processing methods can influence the elastic modulus of human amniotic membranes. For example, fresh, dried, and crosslinked membranes all have slight elastic modulus variances, further aiding the point that not all amniotic membranes are “the same” [53,54].

Viscoelasticity, another key mechanical property, describes the time-dependent response of amniotic membrane to mechanical loading. This property is important for maintaining tissue integrity and function under physiological conditions. The thickness and porosity of amniotic tissues also influence its mechanical properties, with thicker membranes generally exhibiting higher tensile strength but lower flexibility. Depending on the clinical location that an amniotic membrane is utilized, understanding the viscoelastic properties of the material allows clinicians to make an optimal product selection, aiding in the best potential clinical outcome for the patient.

Overall, these initial biological and mechanical studies demonstrate the unique properties of Axolotl DualGraft™; however, it only represents a standalone assessment. Comparative analysis with other similar products on the market remains a critical area of exploration. The literature lacks comprehensive studies comparing the mechanical and biological properties of amnion–amnion and amnion–chorion products. Future research should address this gap by evaluating the suture strength and other mechanical properties of dHAAM against those of skin substitutes and comparable products. Given that the dHAAM used in this study is not decellularized during processing, aids in an additional area of potential exploration compared to amniotic membranes that are decellularized. Further antimicrobial characterization using more sensitive methods could provide a clearer picture of dHAAM’s potential clinical applications.

## 5. Conclusions

While the current study has provided important baseline data on the mechanical and biological properties of dHAAM, further research is necessary to fully understand the implications of these findings. The potential of amniotic membrane HCT/Ps in clinical applications, particularly in surgical applications, wound healing, and tissue repair, remains promising, provided that future studies address the biologic and processing variability and establish standardized protocols for assessment and use.

## Figures and Tables

**Figure 1 bioengineering-11-00953-f001:**
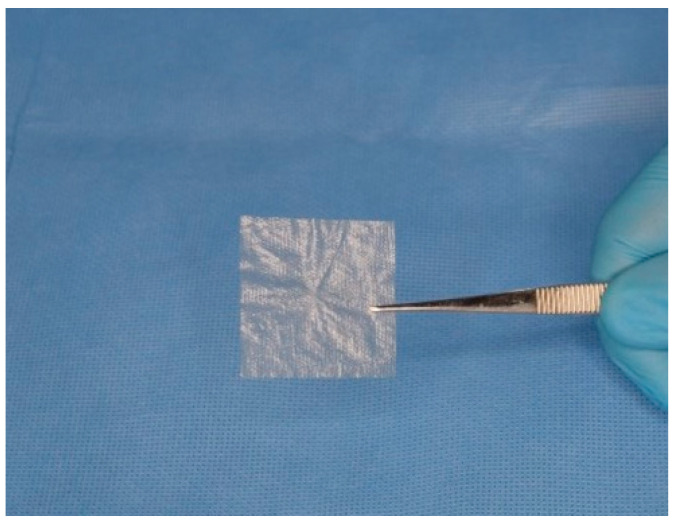
Demonstration of Axolotl DualGraft™ (dHAAM): Sample of dehydrated human amnion–amnion membrane held by sterile tweezers.

**Figure 2 bioengineering-11-00953-f002:**
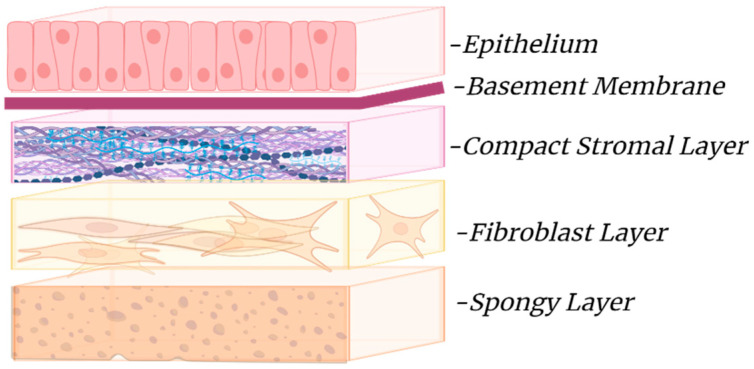
Basic structure of the amniotic membrane: The amniotic membrane is composed of the epithelial layer (epithelium), the basement membrane, and the stroma layer, which is made up of three layers; an inner compact stromal layer, a middle fibroblast layer, and the outermost spongy layer (BioRender.com 2024).

**Figure 3 bioengineering-11-00953-f003:**
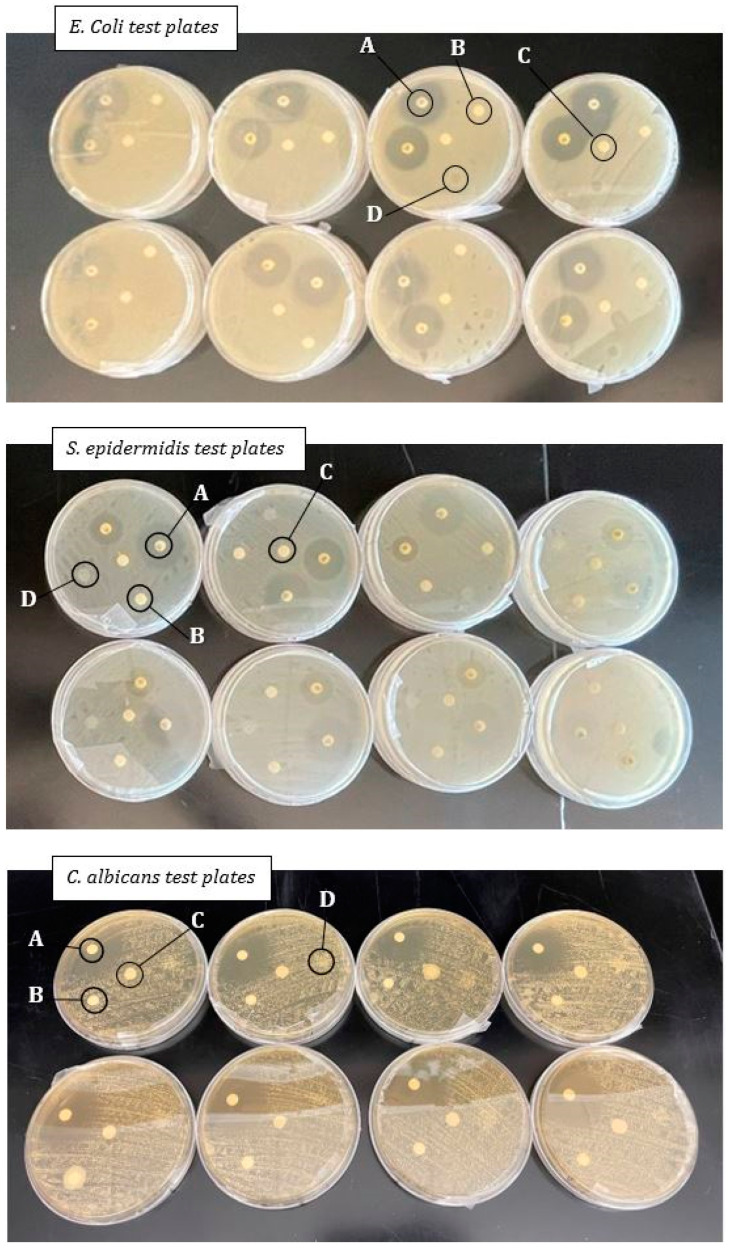
Antimicrobial test plate setup: Disks located in spot A are the antibiotic disks, those in spot B are the dHAAM samples, spot C is the control disk, and spot D is the disk treated with cell conditioned media. *E. coli* plates are the top collection of plates and the *S. epidermidis* plates are the middle, both photographed at the 16 h mark. *C. albicans* is the bottom photo, taken at the 24 h mark. Zones of inhibition, if present, are noted as the clear areas around each disk (Kovacs Z. (2022), A Pilot Study for Antimicrobial Characterization of Two Axolotl Products Using Disk Diffusion. Internal Axolotl Biologix report: unpublished).

**Figure 4 bioengineering-11-00953-f004:**
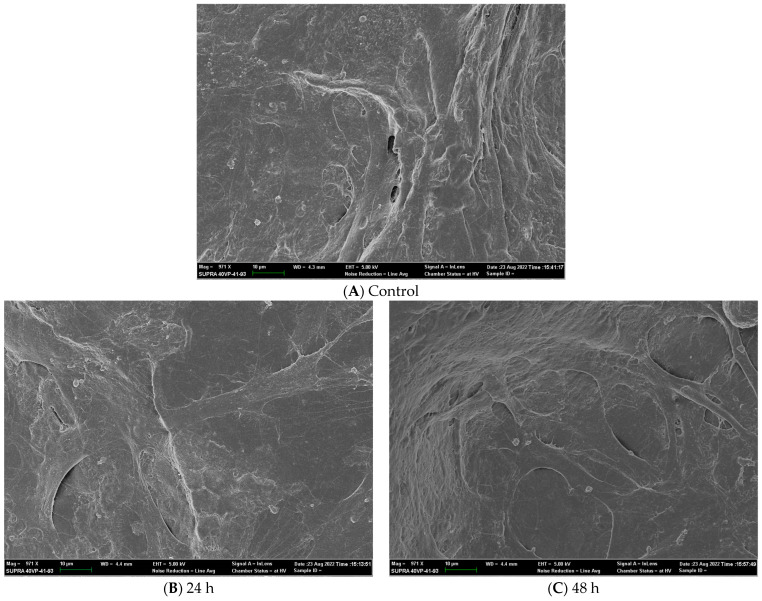
SEM images of adult HDF cells on Axolotl Biologix DualGraft^TM^: (**A**) Control SEM image is from a membrane sample fixed after 48 h incubation in DMEM; mag 971×. (**B**) Adult HDF cells seeded and incubated for 24 h on membrane sample before being fixed; mag 971×. (**C**) Adult HDF cells seeded and incubated for 48 h on membrane sample before being fixed; mag 971× (we used BioRender.com 2024 to combine images into one unit; images are from Ingraldi, A. (2023) DualGraft with HDF SEM Report. Internal Axolotl Biologix report: unpublished).

**Figure 5 bioengineering-11-00953-f005:**
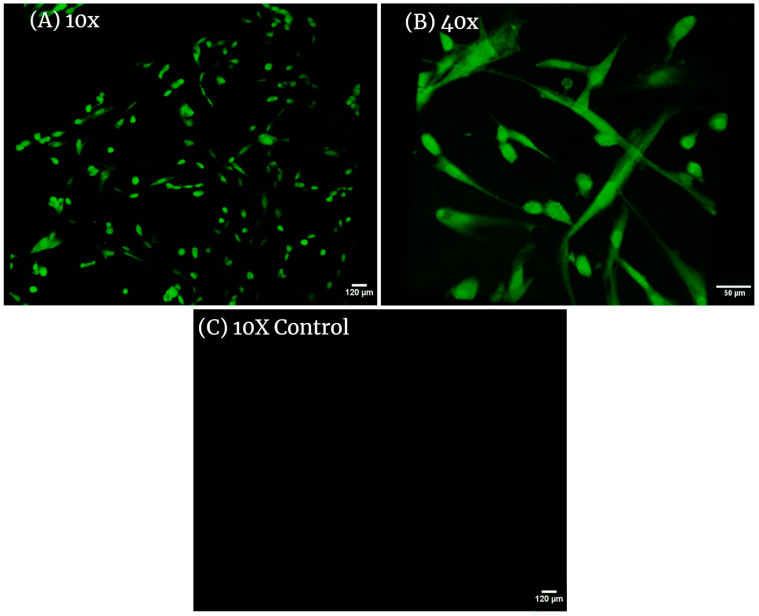
Confocal images of adult HDF cells on Axolotl Biologix DualGraft^TM^: (**A**) 10× magnification in confocal microscopy (Leica TCS SPE II) with 488 nm laser shows many cells present within the membrane; this image was taken using the Z-stack feature of the confocal micropscope, outlining the 3D placement of cells within the membrane; (**B**) 40× magnification presents hDF with normal morphology with elongated arms and rounded cell bodies either recently attached or preparing for division; (**C**) 10× magnification of unseeded membranes treated with fluorescent probe (control sample); no cells are highlighted, demonstrating that the membrane is decellularized and cells viewed in seeded samples are true positives. We used BioRender.com 2024 to combine images into one unit; images are from Ingraldi, A. (2023) Cell Tracker Green HDF on Membrane Report. Internal Axolotl Biologix report: unpublished.

**Figure 6 bioengineering-11-00953-f006:**
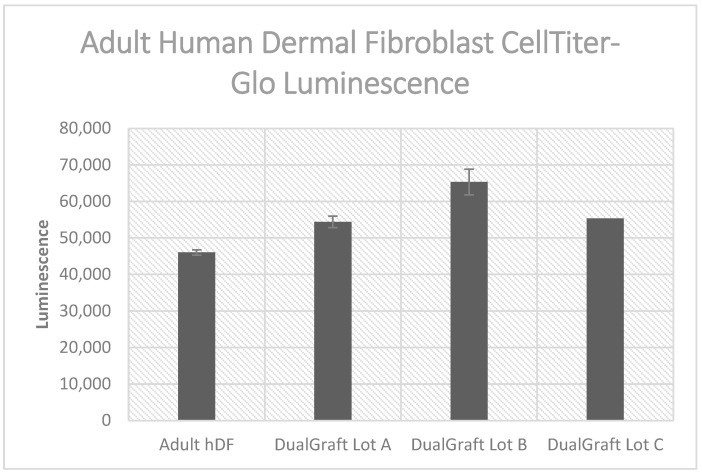
Luminescent cell viability assay results: Axolotl DualGraft™ samples (Lot A–C, luminescence > 54,300) presented a greater luminescence than adult human dermal fibroblasts cells (luminescence < 46,100) cultured without membrane present (Audet R. and Ingraldi A. (2022), DualGraft Biocompatibility—ATP Assay Report. Internal Axolotl Biologix report: unpublished).

**Figure 7 bioengineering-11-00953-f007:**
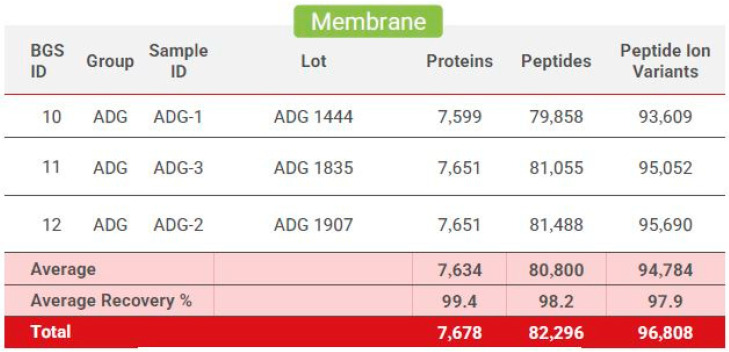
Biognosys proteome protein profile: Generated from three separate dHAAM lots and identifies the total number of proteins, peptides, and peptide ion variants present in each membrane sample (Kamber D. and Soste M. (2022) Proteomic Analysis of Axolotl Ambient and Axolotl DualGraft—Final Report: unpublished).

**Figure 8 bioengineering-11-00953-f008:**
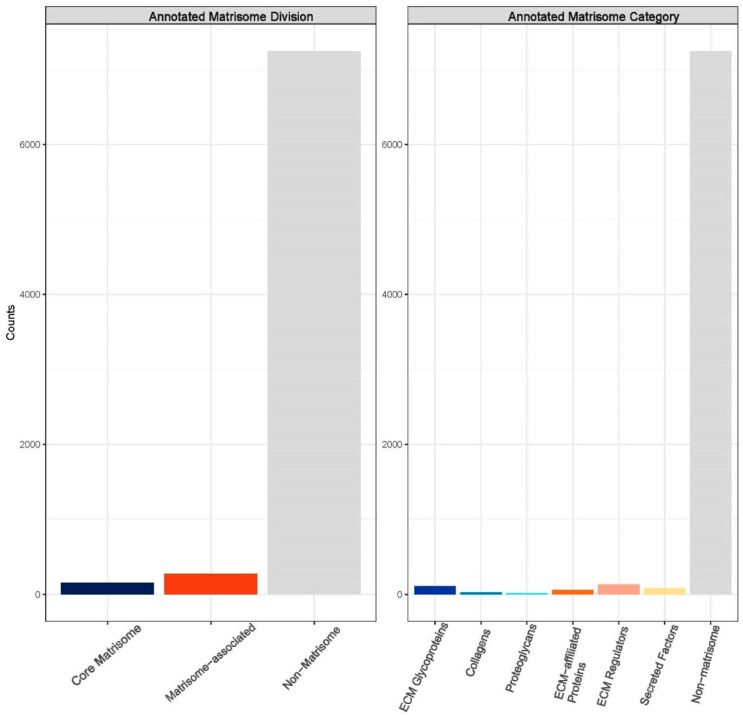
Annotated matrisome divisions: The matrisome AnalyzeR identified a total of 163 core matrisome proteins, a total of 290 matrisome-associated proteins, and a total of 7228 non-matrisome-associated proteins. (Audet R. and Ingraldi A. (2024) Matrisome AnalyzeR DualGraft Report. Internal Axolotl Biologix report: unpublished).

**Figure 9 bioengineering-11-00953-f009:**
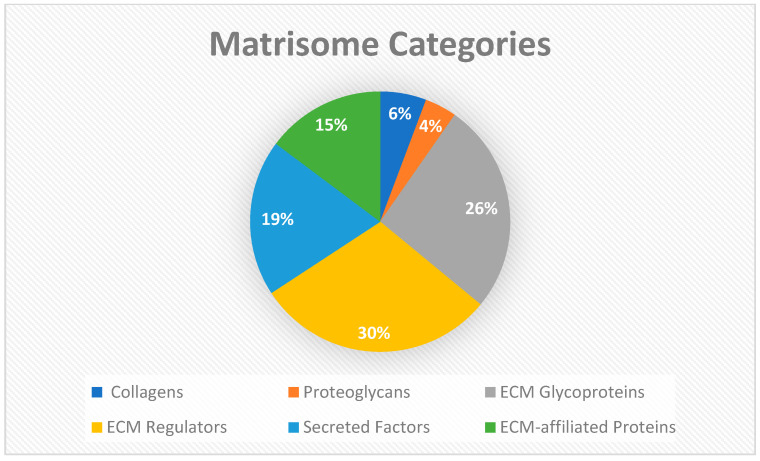
Matrisome categories: The matrisome AnalyzeR identified a total of 26 collagens, 18 proteoglycans, and 119 ECM glycoprotein genes for total of 163 core matrisome genes. A total of 135 ECM regulators, 88 secreted factors, and 67 ECM-affiliated proteins were identified for a total of 290 matrisome-associated genes. (Audet R. and Ingraldi A. (2024) Matrisome AnalyzeR DualGraft Report. Internal Axolotl Biologix report: unpublished).

**Figure 10 bioengineering-11-00953-f010:**
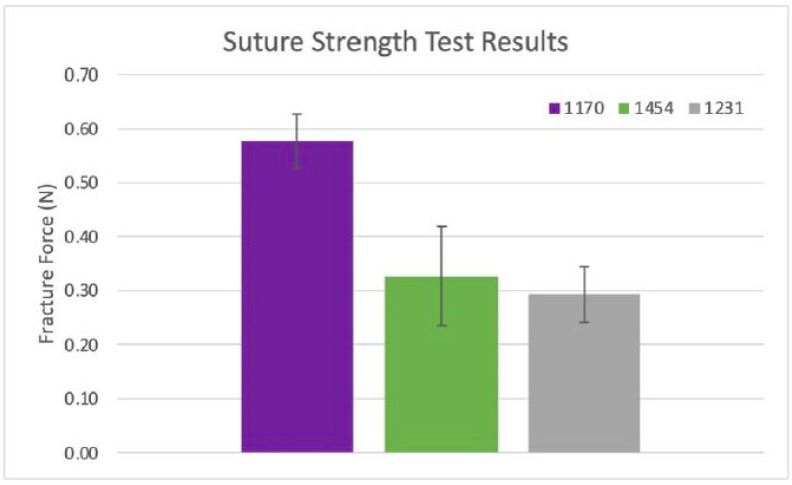
Suture strength results: Ultimate suture strength based on measured forces from two samples per dHAAM donor lot (1170, 1454, 1231). (Becker T., (2023). Suture Strength Testing Report. Internal Axolotl Biologix report: unpublished).

**Figure 11 bioengineering-11-00953-f011:**
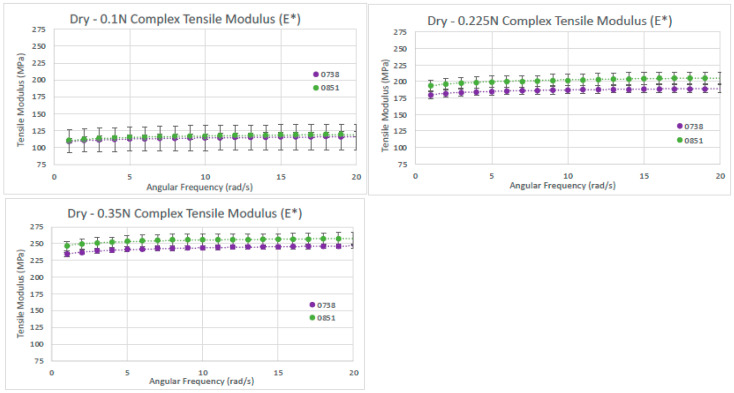
Dry tensile modulus results. Tensile moduli (E*) for two dry dHAAM samples tested at tensile forces of (**left**) 0.1 N, (**right**) 0.225 N, and (lower **left**) 0.35 N (Becker T., (2023). Complex Tensile Modulus Testing Report. Internal Axolotl Biologix report, unpublished).

**Figure 12 bioengineering-11-00953-f012:**
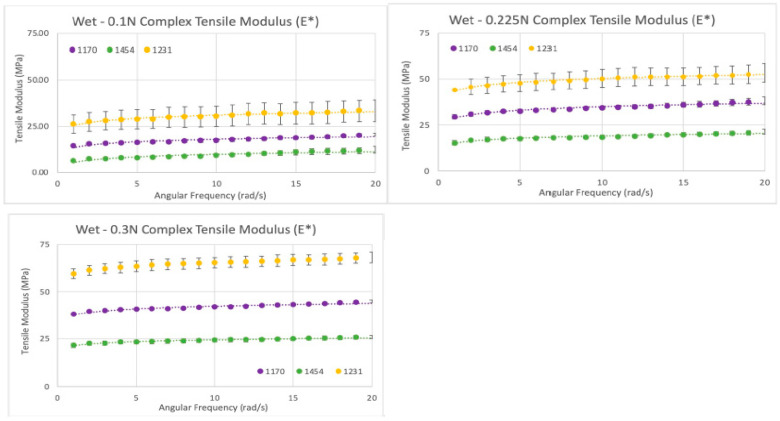
Wet tensile modulus results: Tensile moduli (E*) for wet samples tested at tensile forces of 0.1 N (**top left**), 0.225 N (**top right**), and 0.3 N (**bottom left**) (Becker T., (2023). Complex Tensile Modulus Testing Report. Internal Axolotl Biologix report: unpublished).

**Figure 13 bioengineering-11-00953-f013:**
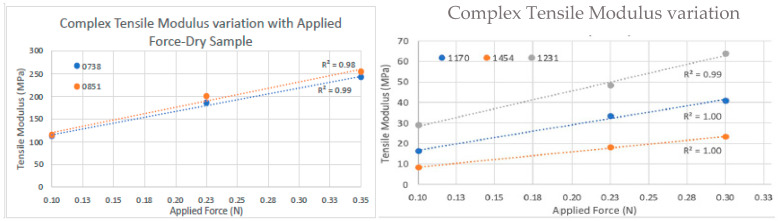
Complex tensile modulus results: (**Left Graph**) Tensile moduli (E.) for dry samples; moduli vs. applied forces (0.1 N, 0.225 N, and 0.35 N). Blue and orange represent the 2 different dHAAM lots tested. (**Right Graph**) Tensile moduli (E) for three lots of samples vs. all applied forces (0.1 N, 0.225 N, and 0.3 N) at 6 rad/s (1 Hz) (Becker T., (2023). Complex Tensile Modulus Testing Report. Internal Axolotl Biologix report: unpublished).

**Figure 14 bioengineering-11-00953-f014:**
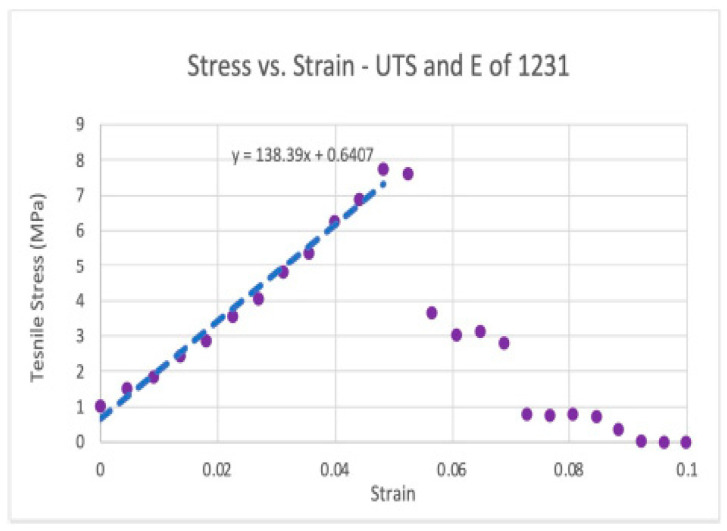
Stress versus strain: Example of tensile stress vs. strain for lot 1231 of dHAAM sample, the slope included = tensile modulus. (Becker T., (2023). Tensile Strength Testing Report. Internal Axolotl Biologix report: unpublished).

**Figure 15 bioengineering-11-00953-f015:**
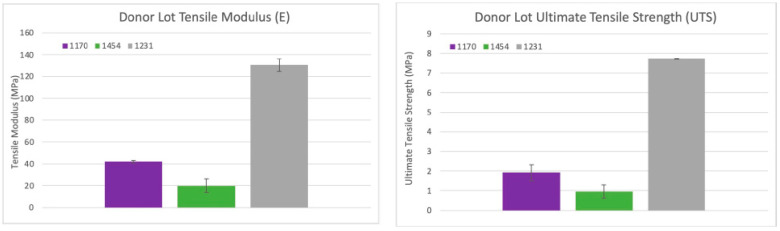
Tensile strength results: Average tensile moduli for the three dHAAM lots tested; ultimate tensile strength for the three donor lots tested. (Becker T., (2023). Tensile Strength Testing Report. Internal Axolotl Biologix report: unpublished).

**Figure 16 bioengineering-11-00953-f016:**
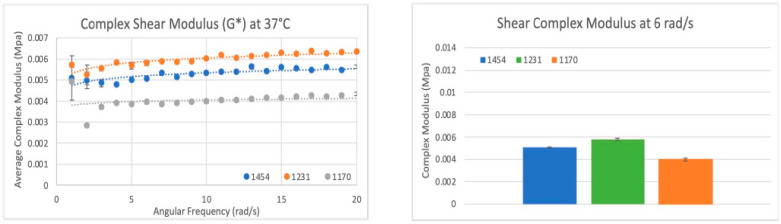
Shear modulus results: (**Left**) Scatter plot of the complex shear modulus (G*) for three sample dHAAM lots tested at 37 °C. (**Right**) Bar graph comparison of shear modulus (G*) for the three samples lots tested at shear rate 6 rad/s (60 BPM); all lot *p* values < 0.01. (Becker T., (2023). Shear Modulus Testing Report. Internal Axolotl Biologix report: unpublished).

**Figure 17 bioengineering-11-00953-f017:**
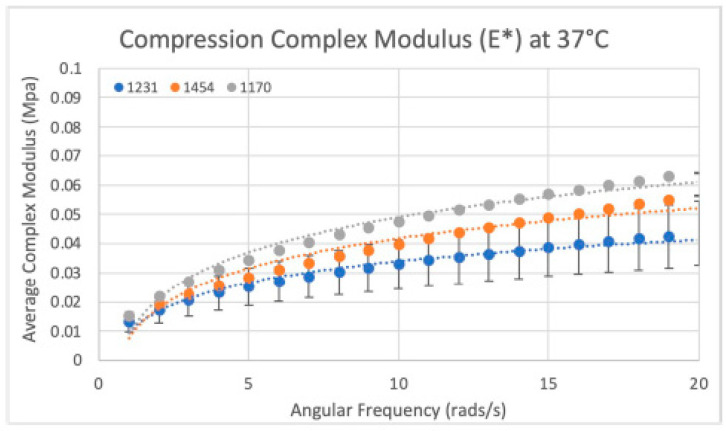
Compressive modulus results: Compressive modulus (E*) for the three dHAAM lots tested at 37 °C. (Becker T., (2023). Compressive Modulus Report. Internal Axolotl Biologix report: unpublished).

**Table 1 bioengineering-11-00953-t001:** Mean Inhibitory Zone: The average zone of inhibition for each antibiotic treatment per organism in mm (n = 8). The dHAAM and conditioned media samples performed similarly to the control group, overgrown with the organisms on the plate, resulting in no inhibition.

Mean of Inhibitory Zone (n = 8)							
Organism	Tetracycline	Chloramphenicol	Gentamycin	Fluconazole	DualGraft	Cond. Media	Control
*E. coli*	26	27	-	-	0	0	0
*S. epidermidis*	18	-	26	-	0	0	0
*C. albicans*	-	-	-	30	0	0	0

**Table 2 bioengineering-11-00953-t002:** Ultimate suture strength: Ultimate suture strength (USS) determination and suture force based on suture size. (Becker T., (2023). Suture Strength Testing Report. Internal Axolotl Biologix report: unpublished).

Ultimate Suture Strength (USS) Table				
Suture Size (USP)	Suture Diameter (mm)	Suture Area (mm^2^)	USS (MPa)	Suture Force (N)
6-0	0.07	0.005	8.5	0.05
5-0	0.10	0.008	8.5	0.07
4-0	0.15	0.012	8.5	0.10
3-0	0.20	0.016	8.5	0.13
2-0	0.30	0.024	8.5	0.20
0	0.35	0.027	8.5	0.23

**Table 3 bioengineering-11-00953-t003:** Category—core matrisome (total identified = 163).

Annotated Gene	Protein Name	Protein Description	Protein Function
DCN	PGS2_HUMAN	Decorin	A small leucine-rich proteoglycan involved in the regulation of collagen fibrillogenesis and matrix organization.
COL1A2	CO1A2_HUMAN	Collagen alpha 2(I) chain	Part of type I collagen, which is a major structural protein in connective tissues.
COL1A1	CO1A1_HUMAN	Collagen alpha 1(I) chain	Also a component of type I collagen, working together with the alpha 2 chain to form the collagen fibrils that provide structural support and strength to various tissues.
LUM	LUM_HUMAN	Lumican	A member of the small leucine-rich proteoglycan family, lumican plays a role in collagen fibril organization.
OGN	MIME_HUMAN	Mimecan	Mimecan is a small leucine-rich proteoglycan involved in regulating collagen fibrillogenesis and contributing to bone and cartilage matrix organization.
PRELP	PRELP_HUMAN	Prolargin	A small leucine-rich proteoglycan that influences collagen fibril formation and tissue repair, playing a role in the structural integrity of connective tissues.
DSP	DESP_HUMAN	Desmoplakin	A key component of desmosomes, which are cell structures involved in maintaining cell–cell adhesion and structural integrity in tissues such as the skin and heart.
FN1	FINC_HUMAN	Fibronectin	A glycoprotein which plays a crucial role in cell adhesion, growth, migration, and wound healing. It helps to organize the extracellular matrix and facilitate cellular interactions with the matrix.
COL3A1	CO3A1_HUMAN	Collagen alpha 1(III) chain	Part of type III collagen, which is found in many tissues including skin, blood vessels, and internal organs. Type III collagen provides structural support and flexibility.
LTBP4	LTBP4_HUMAN	Latent-transforming growth factor beta-binding protein 4	A protein that binds to and regulates the activation of transforming growth factor beta (TGF-β), which is involved in cell growth, differentiation, and extracellular matrix production.

**Table 4 bioengineering-11-00953-t004:** Category—ECM glycoproteins (total identified = 119).

Annotated Gene	Protein Name	Protein Description	Protein Function
DSP	DESP_HUMAN	Desmoplakin	Component of desmosomes, which are adhesive junctions that provide mechanical strength to tissues by linking intermediate filaments of the cytoskeleton to the cell membrane, thereby maintaining cell–cell adhesion.
FN1	FINC_HUMAN	Fibronectin	Fibronectin is a multifunctional glycoprotein involved in cell adhesion, migration, and matrix organization. It plays a key role in wound healing and tissue repair by facilitating interactions between cells and the extracellular matrix.
LTBP4	LTBP4_HUMAN	Latent-transforming growth factor beta-binding protein 4	Binds to and regulates the activation of transforming growth factor beta (TGF-β), influencing cell growth, differentiation, and extracellular matrix production.
TGFBI	BGH3_HUMAN	Transforming growth factor-beta-induced protein ig-h3	Induced by TGF-β and involved in cell adhesion, migration, and extracellular matrix organization. It plays a role in tissue repair and fibrosis.
THBS1	TSP1_HUMAN	Thrombospondin-1	A glycoprotein involved in cell–cell and cell–matrix interactions, thrombospondin-1 regulates processes such as angiogenesis, wound healing, and tissue remodeling. It can influence cell adhesion and migration by interacting with various cell surface receptors.
TNXB	TENX_HUMAN	Tenascin-X	A large extracellular matrix glycoprotein that affects collagen fibril organization and tissue elasticity. It is involved in connective tissue structure and has roles in tissue repair and development.
FBN1	FBN1_HUMAN	Fibrillin-1	A key component of microfibrils in the extracellular matrix, fibrillin-1 provides structural support and elasticity to connective tissues. It is crucial for the integrity of tissues such as skin, lungs, and blood vessels.
CRISPLD2	CRLD2_HUMAN	Cysteine-rich secretory protein LCCL domain-containing 2	This protein is involved in cellular processes such as adhesion and migration.
NID1	CO3A1_HUMAN	Nidogen-1	Nidogen-1 is a glycoprotein that links laminin and collagen IV in the basement membrane. It plays a critical role in basement membrane stability and cell–matrix interactions.
ABI3BP	LTBP4_HUMAN	Target of Nesh-SH3	This protein interacts with various cytoskeletal and signaling proteins, influencing cell adhesion, migration, and the cytoskeleton’s organization. It is involved in processes such as cell motility and signal transduction.

**Table 5 bioengineering-11-00953-t005:** Category—collagens (total identified = 27).

Annotated Gene	Protein Name	Protein Description	Protein Function
COL1A2	CO1A2_HUMAN	Collagen alpha 2(I) chain	Part of type I collagen, this chain, along with the alpha 1(I) chain, forms type I collagen fibrils. Type I collagen provides tensile strength and structural support to connective tissues such as skin, tendons, and bones.
COL1A1	CO1A1_HUMAN	Collagen alpha 1(I) chain	This is another component of type I collagen, working with the alpha 2(I) chain to form type I collagen fibrils that provide structural integrity and strength to various connective tissues.
COL3A1	CO3A1_HUMAN	Collagen alpha 1(III) chain	Part of type III collagen, this chain combines with the alpha 1(III) chain to form type III collagen, which provides structural support and flexibility to tissues such as skin, blood vessels, and internal organs.
COL2A1	CO2A1_HUMAN	Collagen alpha 1(II) chain	This chain is a component of type II collagen, which is primarily found in cartilage. Type II collagen provides tensile strength and elasticity to cartilage, essential for joint function and support.
COL6A3	CO6A3_HUMAN	Collagen alpha 3(VI) chain	Part of type VI collagen, this chain contributes to the formation of collagen VI fibrils, which are important for anchoring and organizing other matrix proteins and maintaining the structural integrity of tissues such as muscle and skin.
COL5A2	CO5A2_HUMAN	Collagen alpha 2(V) chain	This chain is part of type V collagen, which works with type I and type III collagens to regulate fibril diameter and organization, influencing tissue strength and flexibility.
COL17A1	COHA1_HUMAN	Collagen alpha 1(XVII) chain	A component of type XVII collagen, also known as BP180, which is a key component of hemidesmosomes, important for the adhesion of the epidermis to the underlying dermis, providing stability to the skin.
COL7A1	CO7A1_HUMAN	Collagen alpha 1(VII) chain	This chain is part of type VII collagen, which forms anchoring fibrils that connect the epidermis to the dermis, contributing to skin stability.
COL6A1	CO6A1_HUMAN	Collagen alpha 1(VI) chain	This chain is a component of type VI collagen, which forms microfibrils that support and stabilize the extracellular matrix and connect to other collagen types, influencing tissue integrity and elasticity.
COL5A1	CO5A1_HUMAN	Collagen alpha 1(V) chain	A part of type V collagen, which works with other collagen types to regulate collagen fibril formation and organization, affecting tissue structure and function.

**Table 6 bioengineering-11-00953-t006:** Category—proteoglycans (total identified = 18).

Annotated Gene	Protein Name	Protein Description	Protein Function
DCN	PGS2_HUMAN	Decorin	A small leucine-rich proteoglycan involved in collagen fibril formation and matrix organization. It binds to collagen, modulating its assembly and stability, and plays a role in tissue repair and fibrosis.
LUM	LUM_HUMAN	Lumican	A member of the small leucine-rich proteoglycan family, lumican is involved in collagen fibril organization and contributes to corneal transparency and tissue repair. It affects the alignment of collagen fibers in connective tissues.
OGN	MIME_HUMAN	Mimecan	Also known as osteoglycin, mimecan is a small leucine-rich proteoglycan that regulates collagen fibrillogenesis and contributes to bone and cartilage matrix organization. It plays a role in tissue repair and the structural integrity of connective tissues.
PRELP	PRELP_HUMAN	Prolargin	A small leucine-rich proteoglycan that influences collagen fibril formation and tissue repair. It is involved in maintaining the structural integrity of connective tissues by interacting with other matrix proteins.
BGN	PGS1_HUMAN	Biglycan	A small leucine-rich proteoglycan involved in regulating collagen fibril formation and matrix organization. It binds to various matrix proteins and growth factors, influencing tissue repair and cellular processes.
VCAN	CSPG2_HUMAN	Versican core protein	A large chondroitin sulfate proteoglycan that is important for cell adhesion, migration, and tissue hydration. Versican plays a role in extracellular matrix organization and can influence tissue development and repair.
OMD	OMD_HUMAN	Osteomodulin	A small leucine-rich proteoglycan involved in bone matrix organization. It regulates collagen fibril assembly and mineralization, contributing to bone strength and structure.
HSPG2	PGBM_HUMAN	Basement membrane-specific heparan sulfate proteoglycan core protein	A key component of basement membranes, this proteoglycan binds to various matrix proteins and growth factors, influencing cell adhesion, proliferation, and matrix organization.
PRG2	PRG2_HUMAN	Bone marrow proteoglycan	Also known as osteoglycan, it is involved in bone matrix organization and mineralization, playing a role in bone development and repair.
ASPN	ASPN_HUMAN	Asporin	A small leucine-rich proteoglycan that modulates collagen fibril formation and influences tissue repair. It is involved in the regulation of matrix organization and cellular interactions within connective tissues.

**Table 7 bioengineering-11-00953-t007:** Category—ECM regulators (total identified = 135).

Annotated Gene	Protein Name	Protein Description	Protein Function
CTSD	CATD_HUMAN	Cathepsin D	A lysosomal aspartic protease is involved in the degradation of proteins within lysosomes. It plays a role in various cellular processes, including antigen processing, apoptosis, and tissue remodeling
A2ML1	A2ML1_HUMAN	Alpha-2-macroglobulin-like protein 1	This protein is part of the alpha-2-macroglobulin family, which functions as a broad-spectrum protease inhibitor. It inhibits a wide range of proteases, playing a role in regulating proteinase activity and modulating inflammatory responses
PLG	PLMN_HUMAN	Plasminogen	The precursor of plasmin, a protease involved in fibrinolysis. Plasminogen is converted to plasmin, which breaks down fibrin in blood clots, playing a critical role in the regulation of blood clotting and tissue remodeling.
CTSB	CATB_HUMAN	Cathepsin B	A lysosomal cysteine protease is involved in the degradation of proteins and peptides. It participates in various processes including protein turnover, antigen processing, and tissue remodeling. Cathepsin B is also implicated in certain pathological conditions such as cancer.
PLOD3	PLOD3_HUMAN	Multifunctional procollagen lysine hydroxylase and glycosyltransferase LH3	An enzyme involved in collagen biosynthesis. It performs lysyl hydroxylation and glycosylation of collagen, which are critical for the stability and function of collagen fibers.
SERPINH1	SERPH_HUMAN	Serpin H1	This protein is a molecular chaperone specific for collagen. It is involved in the proper folding and assembly of collagen molecules in the endoplasmic reticulum.
P4HA1	P4HA1_HUMAN	Prolyl 4-hydroxylase subunit alpha-1	A component of the prolyl 4-hydroxylase enzyme complex, which is essential for collagen synthesis. It hydroxylates proline residues in collagen, a modification crucial for collagen stability and function.
SERPINB2	PAI2_HUMAN	Plasminogen activator inhibitor 2	A protein that inhibits tissue plasminogen activator (tPA) and urokinase-type plasminogen activator (uPA), thereby regulating fibrinolysis and modulating tissue remodeling and repair processes.
F13A1	F13A_HUMAN	Coagulation factor XIII A chain	A subunit of factor XIII, which is involved in blood clot stabilization. It crosslinks fibrin polymers, strengthening and stabilizing blood clots to prevent excessive bleeding.
PLOD1	PLOD1_HUMAN	Procollagen-lysine,2-oxoglutarate 5-dioxygenase 1	An enzyme involved in the hydroxylation of lysine residues in collagen precursors. This modification is important for the formation of stable collagen fibers and proper collagen matrix assembly.

## Data Availability

The datasets presented in this article are not readily available because corporate policy and an ongoing study with dHAAM. Requests to access the datasets should be directed to Aaron J. Tabor at atabor@axobio.com.

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
