# Peer review of "Characterization of Amnion-Derived Membrane for Clinical Wound Applications"

_bioengineering, 2024, doi:10.3390/bioengineering11100953_

Round 1
Reviewer 1 Report
Comments and Suggestions for Authors
This paper aims to evaluate the cellular biocompatibility and mechanical properties of the Axolotl Biologix DualGraft™ membrane product to better understand its potential as a skin substitute.
Major Comments:
1. Please rewrite the introduction concisely, adhering to the journal's format. Avoid using paragraphs.
2. Remove these figures (Figure 1-5) from the text.
3. This section is overly detailed and lengthy. Please rewrite it in accordance with the journal's full article format.
4. Revise all figure captions for clarity and readability, as they are currently difficult to follow.
5. All results sections lack a control group.
6. The paper lacks consistency and logical flow, making it difficult to read. Additionally, the overall writing format does not align with scientific paper standards.
7. Authors should review journal articles in this field to ensure that the writing style and structure align with scientific standards.
Specific Comments: Biological assay
1. Table I: Please remove this table as it does not need to be included in the text.
2. Figure 6: This figure lacks a control group (dHAAM alone). The data are negative, as the properties of dHAAM did not diffuse into the agarose or provide a zone of inhibition. The experiment should be repeated. Avoid including literature discussion in the results; if included, it should adhere to the journal's format.
3. Figure 7: This figure also lacks a control group (dHAAM alone). For negative data, if differences are observed, the experiment should be repeated. Note that this experiment is a cell attachment study, not a biocompatibility test.
4. Figure 8: This figure lacks a control group. Additionally:
Why was the test conducted for 72 hours instead of 24 or 48 hours? The study results indicate cell attachment growth effects rather than cytocompatibility testing.
5. Figure 9: Refrain from including descriptions of experimental reagents (e.g., CellTiter-Glo® Luminescent Cell Viability Assay from Promega) in the research results. Ensure the format follows the journal's guidelines.

none
Reviewer 2 Report
Comments and Suggestions for Authors
Review of the Publication: "Characterization of Amnion-Derived Membrane for Clinical Wound Applications"
The article addresses a very interesting research topic; however, several questions and comments have arisen during the review process, which I present below:
- Abstract: The abstract is incorrectly prepared—it's too general and lacks results and conclusions. The general part of the abstract should be about 1/3, while 2/3 should include numerical data from the key findings and 1 to 2 conclusions drawn from the research.
- Introduction: Since this is a research paper and not a review, the introduction is too lengthy. It should be no more than 1.5 pages. It should refer to as many recent literature sources as possible, highlight the research objective, identify the research gap, and describe the theoretical background related to the topic. The introduction should discuss aspects investigated by other researchers and point out the research gap that the presented results will fill. Therefore, please completely revise and shorten the introduction. It should be a concise text that introduces the reader. The current format distorts the proportions of the article, with a significant emphasis on background information rather than scientific results.
- Section 2.1: There is a lack of reference to Table 1 in the text.
- Methodology: The methodology section must include details about the equipment, time, procedures, pH, reagents, etc. Theoretical explanations, such as: “The biocompatibility of dHAAM used as a wound covering/structural barrier in clinical applications is important because it determines how well the allograft biomaterial interacts with the body and if it is performing its intended function by integrating into the wound bed, protecting from infection, and overall fostering the healing process,” should be avoided. Please clarify this section as you did in Section 2.3 and others.
- Figure Quality: The quality of Figure 3 is very poor—please improve it.
- Line 382: “…rheometer (TA Instruments, New Castle, DE), using a 25 mm x 5 mm tensile clamp apparatus (Figure 2).” – Is this a mistake regarding the reference to Figure 2?
- Duplicate Figures: What is the purpose of including two identical figures (Figures 3 and 4) in the article?
- Line 458: "The zones of inhibition for all antibiotics were in the expected susceptible range for each microorganism (see Table # below)." What does # refer to? The text is quite sloppy in some parts—please review the entire paper before submission.
- Latin Names: Latin names should always be italicized.
- Microscopy Figures: There is a lack of scale bars on the confocal microscopy images.
Overall, the work is very interesting, but in addition to the above-mentioned issues, the results, specifically their graphical representation, require improvement. The authors should review other studies and compare the quality of their results with those of others. The graphs are very unclear and unprofessional. The results are promising and interesting, but the presentation of these results is unacceptable in its current state. Therefore, I request that the graphs be redone using a professional program such as Origin. If these comments are addressed, I will recommend the article for publication. At this moment, I believe that the paper requires significant revisions.
Reviewer 3 Report
Comments and Suggestions for Authors
In this paper, authors explored the Axolotl Biologix DualGraft™ membrane product’s cellular biocompatibility and mechanical properties to understand more about the utility and performance of this specialized skin substitute product.
1. Line 56, it should be Figure 1, not Figure one. Line 362, what it Figure #? Line 382, is it Figure 3 or Figure 2?
2. Tables 1-3 and 6-9 are absence in corresponding text.
3. Figures 7-9 are absence in corresponding text.
4. Lines 247-248, what is the unit mcg?
5. E. coli, S. epidermidis, and C. albicans should be written with italic.
6. Table 3 is not a normal Table.
7. In Table 4, normal mechanical testing data such as tensile strength, elongation at break, and Young's modulus are suggested to use.
Round 2
Reviewer 1 Report
Comments and Suggestions for Authors
Authors were adequaltely responded all comments.
Comments on the Quality of English Languagenone
Reviewer 2 Report
Comments and Suggestions for Authors
Thank you for resubmitting the article for review. The authors have put effort into improving it, but it is still not sufficiently well-executed for publication. Below are my comments:
- Please carefully check the MDPI guidelines and verify what should be entered in the corresponding author field (it should be an email address).
- Introduction: Please remove the subsections, i.e., subchapters 1.1, 1.2, 1.2.2, 1.3, and keep the described content as a single section, point 1 only. If the authors cite the guidelines from the MDPI website in response, please also read about the indentation rules in the text—correct this and standardize the format. The text, at first glance, appears unprofessional.
- Citations: In the introduction, the authors cite two references for a single, short sentence, e.g., [3-4]. Please separate them and specify which part of the sentence refers to 3 and which to 4. Additionally, note that in the case of citing references 3 and 4, the correct format should be [3, 4], not [3-4]. If it were [3-5], it would be correct. Each reference the authors cite should be verifiable, meaning it should be cited individually so that both the reviewer and the reader can easily find and verify the reference. The same applies to references 5-8—please separate the references within the sentences into elements that are specifically discussed in those articles. Another example is the reference in line 114—why are the references listed individually? Moreover, citing so many references in a single sentence is incorrect. If the authors want to keep these references, they must describe the information contained in those publications that relate to the topic; otherwise, this might suggest "padding" the literature.
- Formatting note: "Figure 1." (with a period at the end, not a comma)—please check the author guidelines on the MDPI website.
- Lines 60-62: What is the purpose of citing the same reference twice, sentence after sentence?
- Incorrect citation order: The first reference cited is [10], followed by [9]—please correct this.
- Lines 117 to 143: What is the reference here? If it is [28], please note that half a page of text is described for one reference, while earlier, 8 references were cited for a single sentence.
- Line 179: ?
- The section describing the reagents used is missing, i.e., "Materials" (name, company, country of origin, purity, density, etc.).
- In some places, the authors refer to publications by name instead of citing them numerically—the first example I see is in the caption for Figure 3. Is this image from the cited publication?
- Line 466: "Used BioRender.com 2024 to combine images into one unit, images from Ingraldi, A. (2023) DualGraft with HDF SEM Report." Were these images previously published?
- Table 2 is a figure, not a table. Change it to a figure, while Figure 6 is a table. Please ensure the correct labeling.
- What is "Kamber D. & Soste M. (2022) Proteomic Analysis of Axolotl Ambient and Axolotl DualGraft – Final Report"? Has this been previously published? The names given for these authors do not appear in the publication. Why do the authors present this result in the results section as their own?
- Figure 9: Nothing is visible—it lacks quality. I already asked the authors to pay attention to quality, and my comments have still not been addressed.
- Conclusions are missing—there is no such section at all, even though in the MDPI guidelines the authors themselves pasted information that this section is required.
- References are not cited in accordance with MDPI guidelines.
I still have not changed my opinion, and I believe the paper is unprofessional and very chaotic. I suggest thoroughly revisiting it and moving some of the results to supplementary materials. Please propose a new structure for the paper, as the authors are best positioned to decide which parts should go to supplementary materials, and I do not want to impose this on them.
